# Combining Case-Based Reasoning with Deep Learning: Context and Ongoing Case Feature Learning Research

**David Leake, Zachary Wilkerson, David Crandall**

Luddy School, Indiana University, Bloomington, IN 47405, USA
{leake, zachwilk, djcran}@indiana.edu

## Abstract

Case-based reasoning (CBR) and neural network models have complimentary strengths. Neuro-Symbolic hybrids of the two are promising for leveraging the inherently interpretable CBR process with capabilities of neural systems, facilitating processing of nonsymbolic inputs such as images and increasing system accuracy. Such hybrids also can help alleviate challenges for pure neural systems by facilitating explanation of system behavior and integrating domain knowledge. This paper briefly surveys prior directions in combining CBR with deep neural networks and summarizes ongoing research on combining CBR and deep learning to generate indices for case retrieval. This work focuses especially on selecting network architectures and feature extraction locations within a network to improve generation of CBR-useful features.

## Introduction

The success of neural methods, notably learning with deep neural networks, has transformed AI. However, as is well known, such models have limitations as well, such as the need for large sets of training information and the difficulty of explaining their behavior. Much research has focused on addressing explainability as an add-on, with *post-hoc* explanations of deep learning (DL) models, but limitations of such approaches have prompted arguments for using inherently interpretable models (Rudin 2019). The complementary strengths of neural and symbolic models have led to extensive research on combining paradigms to achieve benefits of both (e.g., Hitzler, Sarker, and Eberhart, 2023).

This short paper highlights one area of neurosymbolic integration: combining case-based reasoning (CBR) with learning with deep neural networks. Computer models of CBR are inspired by the human behavior of reasoning by remembering and adapting solutions to prior problems (Leake 1998). CBR systems reason from specific experiences: they solve new problems by retrieving past cases of similar previous problems and adapting the corresponding solution to fit the new problem (Kolodner and Leake 1996). Case-based reasoning offers three main benefits. First, CBR is interpretable; human subjects studies support that presenting retrieved cases is effective for justifying CBR model

decisions (Cunningham, Doyle, and Loughrey 2003; Gates, Leake, and Wilkerson 2023). Second, its ability to analogize from past cases can facilitate solving problems with structured solutions and enable successful reasoning from few examples. Third, it provides multiple opportunities for integrating existing knowledge into the reasoning process, because of the capability to incorporate retrieval knowledge, similarity knowledge, and case adaptation knowledge.

Knowledge used by CBR has traditionally been acquired from domain experts via knowledge engineering. Collecting such knowledge may be difficult or expensive, or it can be infeasible for domains such as image classification. Consequently, integrating CBR with a DL system for tasks such as identifying features for indexing cases could facilitate the application of CBR to nonsymbolic tasks, and the combined systems can provide interpretability and easy knowledge integration, both of which are challenging for DL alone.

DL-CBR integrations have received attention both for improving network performance and explainability (Li et al. 2018; Chen et al. 2019; Barnett et al. 2021; Kenny and Keane 2019; Koch, Zemel, and Salakhutdinov 2015; Vinyals et al. 2016; Sani et al. 2018; Sung et al. 2018; Turner et al. 2018, 2019) and improving CBR system performance (Sani, Wiratunga, and Massie 2017; Ye et al. 2020; Amin et al. 2020; Hoffmann et al. 2020). This short paper highlights some of this foundational work and summarizes current work by the authors on refining DL-based index generation for accurate CBR retrieval (Wilkerson, Leake, and Crandall 2021; Leake, Wilkerson, and Crandall 2022; Leake et al. 2023).

## Case-Based Reasoning

CBR systems generate solutions to new problems by retrieving and adapting the solutions to similar prior problems, and learn by storing the results in a "case base" of prior cases (Kolodner and Leake 1996). Given a new problem, the CBR system retrieves the stored case for the most similar prior problem (e.g., based on weighted Euclidean distance for problems described by a feature vector). It then adapts the prior solution to the differences between the two problems. The adapted solution is applied to the new problem or otherwise evaluated for effectiveness, is annotated with the problem description and result (e.g., was it successful/correct?), and is added to the case base as a new case. CBR systems

perform lazy learning, storing cases without generalization.

## Research Currents for DL-CBR Integrations

DL-CBR integrations have been realized in multiple ways. We examine three directions: applying CBR case-oriented design to make the networks more explainable, learning similarity metrics to improve CBR similarity assessment, and using DL to extract feature information for a CBR system.

### Increasing DL Interpretability and Explaining DL Results

Case-oriented network design has been applied to increase the interpretability of network component/hidden-layer processes (Li et al. 2018; Chen et al. 2019; Barnett et al. 2021). In this work, prototype networks incorporate data sub-component knowledge (e.g., wings, beaks, etc. for bird species classification) directly into the neural architecture as prototype features to encourage learning to align with human-understandable concepts. The networks calculate similarity scores between outputs from one hidden layer and the set of prototypes, which in turn are passed into the later layers that inform the classification. Such models require knowledge that other DL models do not, but they make decisions on a smaller set of more explainable features.

CBR has also been used to provide *post-hoc* explanations of otherwise black-box DL systems (Bach and Mork 2020; Kenny and Keane 2019). DL-CBR "twin systems" combine DL networks with CBR models applying the same feature data, to enable explanation of network decisions based on cases that are similar according to factors important to the network output (Kenny and Keane 2019).

### Using Networks for CBR Similarity Assessment

Neural metric learners predict the likelihood that two examples belong to the same class. Approaches include Siamese networks (Koch, Zemel, and Salakhutdinov 2015), relational networks (Sung et al. 2018), and matching networks (Vinyals et al. 2016). CBR systems can leverage such approaches for similarity calculations using neural models instead of traditional methods (Mathisen et al. 2020). Such methods have been applied to integrations in a variety of domains (e.g., natural language processing (Amin et al. 2020) and workflow processing (Hoffmann et al. 2020)); additionally, using neural systems both for similarity and case adaptation enables both methods to be tuned in combination to maximize overall performance (Ye et al. 2020).

### DL Feature Extraction for CBR

Many CBR systems use feature-vector problem representations for similarity calculations (e.g., via Euclidean distance between vectorized feature sets), with features determined by a domain expert. Feature extraction from the outputs of DL hidden layers is appealing as an alternative method to augment existing domain knowledge (e.g., (Turner et al. 2018, 2019; Sani, Wiratunga, and Massie 2017)), including for generating features for hard-to-analyze data such as images. Such feature extraction models leverage the effectiveness of DL models for condensing multi-dimensional input

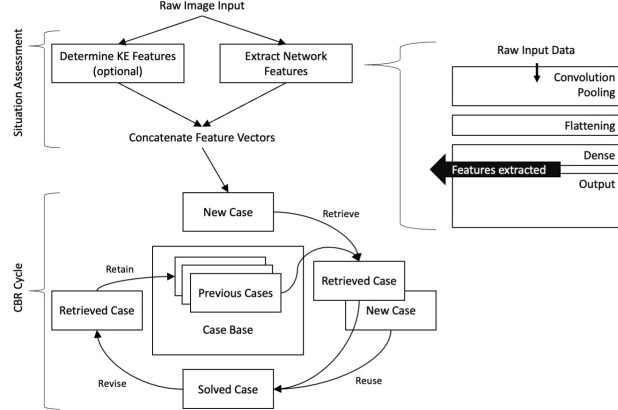

Figure 1: Feature extraction dataflow from the DL model (right) to the CBR cycle (bottom left, after Aamodt and Plaza (1994)). In our approach, features are extracted prior to the output layer of the DL model and combined with any existing knowledge-engineered (KE) features to form a feature vector for CBR retrieval. The figure illustrates feature extraction from a convolutional neural network structure but may be generalized to other DL models from which feature vectors may be extracted. Figure is from Leake et al. (2023).

data into feature vectors for CBR retrieval. This can inform relative classification on example images to support a DL classifier when it is not confident in its decision, by enabling the system to group a query with other similar examples in the case base (Turner et al. 2018, 2019). Additionally, nearest-neighbor classification with features extracted from a network can outperform handcrafted features in domains with complex feature data (e.g., Sani, Wiratunga, and Massie, 2017).

## Recent Research on Extraction of DL Network Features to Use for CBR Retrieval

Recent research develops methods for extracting features from deep neural networks to use for case retrieval, with the goal of generating features that result in higher CBR system accuracy for a classification task. The aim is to exploit neurally-generated information to maximize CBR performance in order to provide effective and interpretable classification. Figure 1 illustrates the extraction dataflow. Our work studies how the quality of DL-extracted features for CBR is influenced by factors such as the DL model architecture and the layers from which features are extracted.

Many factors affect the interplay between DL and CBR systems combined for DL-CBR integrations. One is training procedure. Many DL-CBR systems train the DL model independently and use the CBR system for testing; this reduces training cost by avoiding repeated CBR retrievals during training, but systems that consider CBR performance during training may have the potential to generate higher-quality features. We plan to explore this in our future work.

Feature quality could also be influenced by DL system parameters, either at the model level (e.g., architecture choice,

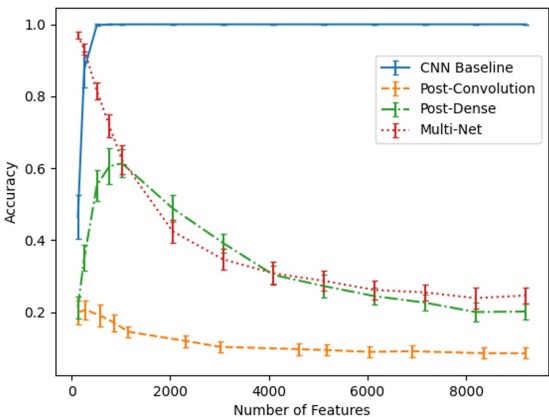

Figure 2: Classification accuracy by number of features extracted, for the Animals with Attributes data set (Xian et al. 2018). Each series shows a different feature extraction location (i.e., after convolution layers or densely-connected layers) or structure (i.e., Multi-net), compared against the baseline network model. Figure from Leake, Wilkerson, and Crandall (2022).

feature extraction location, etc.) or the layer level (e.g., number of features extracted, activation function, etc.). CBR retrieval is often (but not always) based on comparison of flat feature vectors. Consequently, our current work focuses on extracting flat feature vectors (in contrast to, for example, multi-dimensional feature maps outputted by convolutional layers). Extracting flat feature vectors from convolution or similar layers could be difficult. Overall the range of possible layers and desirability of extraction from other types of layers remains a question for future study.

One of our investigations concerns choice of layers from a CNN from which to extract feature information. Each of the previously described feature extraction models makes the plausible assumption that, when using a convolutional neural network as the feature extractor, features should be extracted immediately after the convolution layers. While this makes sense given the conceptualization of convolution as isolating atomic features (e.g., shapes in images) from multi-dimensional data, newer work hypothesizes that the densely-connected neural network layers combine these atomic features into more complex indices that are more suitable for CBR retrieval (Leake, Wilkerson, and Crandall 2022). This explores the following choices and their effects on feature quality (Figure 2):

1. *Feature extraction location.* We found that extracting features from the penultimate layer (i.e., using the outputs of the layer prior to the output layer) of the network generated higher-quality features, evidenced by higher classification accuracy.

2. *Number of features extracted.* For a given data set, there appears to be an optimal number of features to extract from the network for feature quality; we hypothesize that

using too few hinders DL model convergence, and using too many leads to performance degradation due to the "curse of dimensionality" for nearest-neighbor retrieval.

3. *Multi-net architecture for improved feature quality.* We found that, in contrast to training a single $N$-class classifier, training $N$ binary classifiers to differentiate between examples from each class versus all other classes led to better feature quality. Multi-net also had a smaller optimal number of features extracted, which could help facilitate explanation, at a cost of more expensive training.

## How Network Architecture Affects Feature Quality

Our initial work on feature extraction for CBR retrieval considers only feature extraction from a single DL architecture, AlexNet (Krizhevsky, Sutskever, and Hinton 2012). However, there exist many DL models, including those with classification accuracy superior to AlexNet. Consequently, we tested other models–VGG-19, DenseNet121, and Inception V3 (Khan et al. 2019)—as feature extractors, evaluating their impact on feature quality. We draw the following conclusions from experimental results reported in Leake et al. (2023), as well as preliminary data conducted using pre-trained versions of the same models:

1. *CBR-based classification does not mitigate network overfitting on small-data scenarios.* AlexNet and DenseNet feature extractors appeared most resistent to overfitting in our tests, but the classification accuracy for all models exhibited overfitting for limited data. In some cases, even the training accuracy was low, suggesting the model did not have enough training data for convergence.

2. *Transfer learning can enable high DL-CBR performance.* Preliminary data using pre-trained versions of the same models suggests that accuracy is significantly higher, and that overfitting is less pronounced. In particular, it appears that more complex models (e.g., Inception) produce higher-quality features when pre-trained.

## Using Extracted and Knowledge-Engineered Features in Concert

DL-based feature extraction can produce useful knowledge for CBR systems for domains for which knowledge engineering is costly or inaccurate. However, sometimes useful hand-crafted symbolic indices are already available. We have explored the effects of combining knowledge-engineered features injected with varying degrees of random noise with extracted features for retrieval versus retrieval using either feature set individually (Wilkerson, Leake, and Crandall 2021). The results of these experiments suggest that neural model extraction of features from data can usefully supplement existing knowledge-engineered features, providing the greatest benefit when knowledge-engineered features are potentially inaccurate (Wilkerson, Leake, and Crandall 2021; Leake et al. 2023). In addition, knowledge-engineered features can help offset the suboptimal quality of features extracted from networks that have not converged or that overfit to their training data (Leake et al. 2023).

## Conclusions

This short paper presented a survey of research that combines a symbolic method (CBR) with DL systems for greater performance, flexibility, and explainability. After sketching out some of the landscape of DL-CBR integrations, we described our ongoing research on DL-based feature extraction for CBR retrieval applied to image classification. We believe that opportunities exist to increase feature quality by coupling neural models and CBR more closely, especially through further focus on determining which network structures and locations are most suitable for feature extraction, on the application of knowledge-engineered and extracted features in concert, and integrating CBR system performance criteria into backpropagation training.

## Acknowledgments

This work was funded by the US Department of Defense (Contract W52P1J2093009), and by the Department of the Navy, Office of Naval Research (Award N00014-19-1-2655).

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
