# OpenReview forum: "Combining Case-Based Reasoning with Deep Learning: Context and Ongoing Case Feature Learning Research"
_AAAI.org/2024/Workshop/NuCLeaR — NuCLeaR 2024_

### Official Review · Reviewer_KSQK · 2023-12-06
**The paper focuses on the developmental work on CBR fused with DL with breif background works. It require more concrete evidence in support of the work.**

**Rating:** 4
**Confidence:** 4

**Review:**

The paper focuses on the Case-Based Reasoning (CBR) approach fused with Deep Learning (DL). The authors aim to address the persisting problems such as explainability, performance, and huge training data requirements, of DL approaches by the possibilities in the neuro-symbolic fused approach. The authors provide a brief literature review as a background of their research developments. The paper is well-written and explains the objective clearly. However, the paper needs attention to the following issues.

1. The Title does not match with the focus of the paper. It is stated as a short survey but emphasizes more on the developmental works by the authors. The survey naming is inappropriate as it only serves as a literature review.
2. The background works or literature does not delve deep into justifying the approaches and outcomes.
3. The ongoing research by the author is not explained very clearly and lacks numerical evaluative support. One accuracy graph and architecture diagram is provided. The architecture is not explained clearly and is not self-explanatory.
4. There are no clear takeaway points from the research descriptions and conclusions.

---

### Official Review · Reviewer_MQBc · 2023-12-08
**Good short survey about CBR-DL integration and summary of the authors' current work - Would recommend accepting the paper**

**Rating:** 7
**Confidence:** 3

**Review:**

# Summary

The work presents a comprehensive—within the limited bounds of a brief survey—synthesis and investigation of the integration of Case-Based Reasoning (CBR) with Deep Learning (DL), which aims to integrate the interpretability and knowledge-driven nature of CBR with the powerful feature extraction capabilities of DL models. Starting with a survey that contextualizes the complementary advantages of both artificial intelligence paradigms, the authors systematically delve into various research efforts focused on enhancing DL interpretability with CBR, improving similarity assessments within CBR, and feature extraction using DL for CBR indexing.
The core contribution of the paper is the exploration of neural feature extraction for case retrieval in CBR systems, positing that optimal extraction strategies can substantially increase CBR accuracy.  DL can automate feature extraction, mitigating the challenges of manually acquiring symbolic knowledge, which can be not only resource-intensive, but also prone to inaccuracies.

# Quality and Clarity

The document demonstrates methodological thoroughness in examining the intrinsic strengths of both systems, and suggests that CBR provides inherent interpretability and analogizing capabilities that DL models alone lack. Furthermore, the work clearly elaborates on the advantages of integrating DL into CBR for tasks such as feature extraction for case indexing, suggesting this could enhance CBR applications in domains like image classification. The clarity of the paper is commendable, with clearly defined and well-written sections, and an organized progression from background literature to the authors' current research, facilitated by succinct explanations of complex concepts.

Clarity is only slightly affected by the fact that the "(Anonymous)" citation(s) about the authors' current work cannot be followed to verify the claimed results, but naturally that is not a fault of the authors.
# Originality and Significance

The novelty of this study lies in its proposed Neuro-Symbolic approach to feature extraction that has the potential to augment CBR retrieval accuracy for classification tasks. It is significant in that it addresses a potential gap between symbolic and neural systems—the issue of deep learning's opaqueness and the limited applicability of CBR in non-symbolic domains. By targeting improvements in interpretability without sacrificing neural network accuracy, the paper claims that the current work by the authors positions itself as a meaningful contribution to research in AI hybrid systems.

# Pros & Cons
## Pros:

1. The work presents a possible avenue for integration of DL's data-driven feature extraction capabilities with CBR's interpretability and knowledge integration, something that is becoming increasingly desirable in AI research.
2. The paper presents a short empirical analysis based on different neural network architectures and the effects of feature extraction choices (based on the authors' current work, presumably published in other, anonymized, papers).
3. It proposes a beneficial fusion of knowledge-engineered and DL-extracted features, providing a pathway for synergistic integration of existing domain knowledge with learned representations.

## Cons:

1. It is not immediately clear if the described feature extraction can work on different types of NN layers, or if only specific layers are supported or can even be used (or if even that is desirable). Lacking such details, it is not clear if there is a potential limitation in the scope of DL architectures that can be used, which could affect the generalizability of the presented findings.
2. The paper's depth is restrained by its "brief survey" nature, possibly overlooking the complexities and challenges inherent in the integration of CBR and DL, but again, this is not a fault of the authors'.
3. The impact of the proposed methods on computational resources, such as training time or memory requirements, especially when using different neural architectures, is not clearly articulated.

# (Minor) Errors spotted

- *"First, CBR is interpretable, with human subjects study support for the effectiveness of cases for justifying model decisions (Gates, Leake, and Wilkerson 2023)."* - The sentence needs rewording to make it clearer, as is it's not clear what the point being made is.
- The specific versions of DenseNet and VGG (e.g. 16, 19, or both) being used are not specified.
- The acronyms "CNN" and "k-NN" are not defined before being used in the text - although their meaning is obvious within the context of the paper, they should be defined when first used. Also, it's not immediately obvious what "KE" means in Figure 1, it should be explained on the Figure or in the caption.
- *"…because of the the capability to incorporate retrieval knowledge…"* - The word "the" is repeated.
- *"We draw the following conclusions from experimental results reported in (Anonymous). as well as preliminary data conducted using…"* - A comma instead of a full-stop should be used after the citation.

---

### Decision · Program_Chairs · 2023-12-11

Accept